# Taxonomic Revision of Vampire Moths of the Genus *Calyptra* (Lepidoptera: Erebidae: Calpinae) in Chinese Fauna

**DOI:** 10.3390/insects16050534

**Published:** 2025-05-19

**Authors:** Asad Bashir, Yuqi Cui, Yanling Dong, Zhaofu Yang

**Affiliations:** 1Laboratory of Plant Protection Resources and Pest Management, Ministry of Education, Entomological Museum, College of Plant Protection, Northwest A&F University, Yangling 712100, China; asadbashir@nwafu.edu.cn (A.B.); cuiyuqi4744@nwafu.edu.cn (Y.C.); 2College of Plant Protection, Northwest A&F University, Yangling 712100, China; yanlingd@126.com

**Keywords:** Calpinae, *Calyptra*, Chinese fauna, taxonomy, morphology

## Abstract

Calpinae is a subfamily of nocturnal moths in Erebidae, and this subfamily is well recognized for its unusual feeding habits, which include fruit-piercing and blood-feeding activities that are uncommon in Lepidoptera. This subfamily includes the genus *Calyptra*, commonly known as vampire moths, which is fascinating for its ability to pierce fruit and, in some cases, mammalian skin to feed on blood. These moths are found across Europe, Asia, and parts of Africa, and their feeding habits vary, with some males displaying hematophagy, while others prefer fruit piercing. So far, no thorough research has been conducted on this genus within Chinese fauna, and this study provides a detailed explanation of its taxonomy within this context. In this study, we identified seven species of this genus in Chinese fauna. Illustrations of male and female genitalia are also provided.

## 1. Introduction

The genus *Calyptra* Ochsenheimer, 1816 is an enigmatic group in the family Erebidae and comprises 18 known species [1,2,3] and 1 potential subspecies, primarily distributed across the Old World, with only 1 species, *C. canadensis*, occurring in Canada and the United States. Current classifications assign *Calyptra* and other fruit-piercing genera to the subfamily Calpinae [4,5,6,7,8], which was divided into four tribes, including Anomini Grote 1882, Calpini Boisduval 1840, Phyllodini Hampson 1913, and Scoliopterygini Herrich-Schaffer 1852 [5,6,9,10]. Fibiger and Lafontaine [5] considered *Calyptra* to be 1 of 11 genera within the tribe Calipini.

The genus *Calyptra*, commonly known as vampire moths, is characterized by a modified proboscis with heavily sclerotized barbed hooks to pierce the skin of hard fruits (e.g., peaches and citrus fruits) and, rarely, mammals. In general, *Calyptra* spp. are medium-sized moths in Erebidae with an average wingspan of 36–72 mm. The forewing coloration of most species is varied, from light grayish-brown to dark brown, while the hindwings differ from yellow to brown. Typically, the pigmentation of the head and thorax aligns with the forewings, whereas the abdomen’s color more closely matches that of the hindwings. Within the genus, their antennae vary among species and sexes and exhibit diverse forms, such as unidentate, bidentate, unipectinate, and bipectinate forms [1]. At present, these moths are found in eastern Africa, Europe, the Russian Far East, sub-Himalayan South Asia, and Southeast Asia [3,11,12,13]. In Chinese fauna, 10 species of *Calyptra* have been documented [2,14], including *C. gruesa*, *C. thalictri*, *C. bicolor*, *C. albivirgata, C. lata*, *C. minuticornis*, *C. orthograpta*, *C. hokkaida*, *C. fletcheri*, and *C. parva*.

In biology, the most fascinating aspect of the genus *Calyptra* is their feeding behavior; the males of 10 representative species have been identified as capable of piercing mammalian skin and feeding on blood [11,13]. While these males are facultative blood feeders, there are no recorded instances of females doing the same. Male moths may target mammalian hosts to obtain amino acids or carbohydrates that enhance their fitness; however, blood meals do not seem to extend their lifespan [12]. Hematophagous *Calyptra* also plays a significant role in piercing fruits across South and Southeast Asia [12]. Bänziger [15] presented detailed data on the proboscis length and width, as well as the number and dimensions of tearing hooks and erectile barbs, for *C. eustrigata*, *C. minuticornis*, *C. orthograpta*, and *C. fasciata*. The differences in these features are distinct enough to allow for preliminary identification of these four species based solely on their armature. Nonetheless, he noted that the armature remains fundamentally unchanged and can still penetrate mammalian skin. Two species, *C. thalictri* and *C. fasciata*, exhibit different feeding behaviors depending on their geographic location [11,13]. Furthermore, *C. fletcheri* and *C. thalictri* have exhibited hematophagy only under experimental or semi-experimental conditions [11,13]. The other moth species (*C. bicolor, C. eustrigata*, *C. fasciata*, *C. minuticornis*, *C. ophideroides*, *C. orthograpta*, *C. parva*, and *C. pseudobicolor*) have been observed consuming blood in both natural and laboratory environments.

While exploring the diversity of Erebidae in China, we sorted specimens from various sources which were deposited at Northwest A&F University and other museums and identified seven species of *Calyptra* in China. A key, descriptions, and illustrations of these seven species are provided.

## 2. Materials and Methods

### 2.1. Morphological Study

Species identification was conducted based on external morphological characters and genitalia structures and by following identification keys in previous studies [1,14,16]. Specimens were mainly sourced from the Entomological Museum at Northwest A&F University, Yangling, China. Standard protocols were applied for genitalia extraction and slide preparation [17,18]. Male and female genitalia were immersed in a 10% KOH solution overnight to eliminate soft tissue, subsequently rinsed with distilled water, and examined under a stereo microscope. Dissected genitalia were dehydrated using 50%, 75%, 85%, 95%, and 100% ethanol. This dehydration process was required to remove moisture, which aids in the prevention of tissue distortion, clarity of morphological structures, and long-term storage. To avoid damage or loss of genitalia, all dissected parts were carefully put on microscope slides with Canada balsam as the mounting medium and covered with cover slips. Each slide was labeled with a slide number, gender, and species identification, and the slides were maintained in slide boxes.

Adult specimens were photographed using a Canon digital camera, while genitalia were documented using a Zeiss SteReo Discovery V20 stereomicroscope (Oberkochen, Germany). All image edits were made using Adobe Photoshop 2023 v24.1.1 (Adobe Systems Inc., San Jose, CA, USA). Post-processing consisted of typical scientific illustration processes, such as adjusting exposure, contrast, and color balance, to improve clarity and visual uniformity. Images were also cropped and scaled to provide consistent presentation across the plates. All specimen handling and imaging procedures were conducted in a way to ensure accurate records and comparison of morphological traits.

### 2.2. Occurrence Data and Preparation of Map

The distributional data of *Calyptra* were extracted from voucher specimens and compiled into a CSV file containing species names and geographic coordinates. The data on provincial administrative boundaries and national borders of China were obtained from the website of the National Geomatics Center of China [19]. The distribution of *Calyptra* species in China was generated using ArcGIS Desktop v10.8 (Esri Inc., Redlands, CA, USA) [20] based on the processed distribution point data.

## 3. Results

### 3.1. Taxonomic Overview of the Genus Calyptra

*Calyptra* Ochsenheimer, 1816: 78 [21].

Type species: *Phalaena thalictri* Borkhausen, 1790, subsequently designated by Duponchel 1826, 3 [22].

*Calpe* Treitschke, 1825 [23]. Type species: *Phalaena thalictri* Borkhausen, 1790.

*Hypocalpe* Butler, 1883 [24]. Type species: *Calpe fasciata* Moore, 1882 [25].

*Percalpe* Berio, 1956. Type species: *Calpe canadensis* Bethune, 1865.

**Diagnosis:** The genus *Calyptra* can be identified based on several morphological characteristics, though differentiation from closely related genera may require further examination. Key diagnostic features include the following: (i) forward-projecting, beak-like labial palpi; (ii) ciliate antennae with rami that gradually decrease in length from approximately the mid-point to the apex, or at least to about two-fifths of the total length from the tip; (iii) wingspan of 35 to 72 mm; (iv) the presence of a broadly rounded lobe on the basal half of the forewing’s inner margin, accompanied by a broad excision on the distal half, terminating in a tornus that often forms a sharp hook; (v) outer margin of forewing typically rounded, except in *C. ophideroides*, which exhibits an angular margin (wingspan: 57–72 mm); (vi) a distinct diagonal line extending from the pointed forewing apex toward the middle of the inner margin, though this line may be indistinct or incomplete in *C. fasciata* and *C. imperialis*. The former species is characterized by a prominent dark blotch near the reniform stigma, while the latter is the only African representative (wingspan: 55–62 mm).

**Distribution:** *Calyptra* species are primarily distributed in the Indomalaya region, with some occurring in East Asia. The genus exhibits predominant diversity within the Indomalaya realm, with substantial radiation into East Asia. The geographical distribution of seven *Calyptra* spp. in China included in the present study is shown in Figure 1.

#### 3.1.1. *Calyptra gruesa* (Draudt, 1950)

Figure 2A,B, Figure 3A and Figure 4B.

*Calpe gruesa* Draudt, 1950: 168. Type locality: ‘’Taipei-Shan, West-tien-Shan’’ [Taibai Shan Shaanxi; W. Tianmu Shan, Zhejiang, China].

*Calyptra gruesa* Sugi 1982: 212 [26].

**Material examined:** 22 ♂, **China**, Sichuan, Qingcheng Mountain, 01.VII.2006, Hou Xiaoyan; 1 ♂, Zhejiang, Tianmu Mountain, 02.VII.2005, Zheng Jianwu; 1 ♂, Fujian, Wuyi Mountain, Gua Dun, 02.VII.2006; 1 ♂, Hainan, Ledong County, Jianfengling, 02.VI.2007, Ying Lunqing; 1 ♂, Hainan, Lingshui County, Diaoluo Mountain, 28.V.2007. Li Yankai; 1 ♂, Xinjiang Uygur Autonomous Region, Zhaosu County, Turpan–Khorgas Highway 1300 m, 03.VII.2006. Li Tao, Zhai Qing; 1 ♂, Hubei, Shennongjia Forestry District, 28.VII.2006. Cai Lijun, Zhou Huifeng; 1 ♂, Fujian, Wuyishan City, 19.VII.2006, Peng Lingfei, Yuan Xiangqun; 1 ♂, Hunan, Shimen County, Huping Mountain, 01.VII.2007, Guo Hongdiao; 1 ♂, Hunan, Dadongping, Huping Mountain, 1040 m, 23.VII.2006, Lv Lin, Xiang Guo, Hongwei Yu.

**Diagnosis: Adult.** (**Figure 2A,B**). Forewing length 55–59 mm in males, 58 mm in females. Head and thorax brown with purplish-gray areas. Male antennae bipectinate (0.65–0.7 mm) and characterized by longer outer rami and shorter inner rami. Labial palps brown, with dense hair tufts at apex of second segment concealing third segment. Forewing brown, variably suffused purplish-gray. Basal line dark brown, angled at interior of discal cell, and then slanting basad. Apex acute and slightly produced. Antemedial line straight, dark brown, obliquely incurved. Median line indistinct, dark brown and slightly excurved from costal margin, then recurved posterior to reniform stigma. Prominent reddish-brown diagonal line bordered with dark brown, extending from pointed apex to approximately midpoint of the inner margin, merging at one-third of the distance between lobe and hook. Reniform stigma brownish, vertical bar faint or nearly missing. Outer margin convex medially. Subterminal line dark brown, often indistinct, with a yellowish-brown spot near Cu_1_ vein. Tornal hook distinct. Hindwing brown, darker in terminal area, with a faintly perceptible dark brown outer line and prominent undulating subterminal line. Abdomen brown and fusiform. Androtheca on male tibia II present. Dorsum of male femur I normal.

**Male genitalia:** (**Figure 3A**). Uncus elongated and slightly arcuate. Valvae broad with dense tufts of long setae in median area. Cucullus rounded. Corona with apex extending to cucullus, inward-curved and tapering hook-like at terminus. The saccular process hockey-stick-shaped and slightly tapering, similar to that of *C. albivirgata* but the base relatively enlarged. Juxta dome-shaped. Aedeagus slender and slightly curved, with two patches of densely sclerotized spinous cornuti beyond halfway along the length and fine teeth-like spines near the terminal portion. Additional elongated cornuti structures bearing on the distal end of vesica.

**Female genitalia:** (**Figure 4B**). Papilla analis well developed, sparsely setose. Apophyses slender and relatively shorter than those of congeneric species. Ductus bursae longer than in *C. lata*, with pair of robust sclerites near junction of corpus bursae. Ostium bursae relatively broad, antrum weakly sclerotized. Corpus bursae heavily wrinkled, elongate, and globular with pronounced curvature. Distinct sclerotized hook or ridge of ductus bursae present as a diagnostic feature discriminating from its congeners.

**Host plant:** *Stephania* spp. (Menispermaceae).

**Distribution:** China (Shaanxi, Zhejiang, Hainan, Hubei, Hunan, Fujian, Xinjiang, Sichuan), Japan.

#### 3.1.2. *Calyptra thalictri* (Borkhausen, 1790)

Figure 2C and Figure 3B.

*Phalaena thalictri* Borkhausen, 1790: 425. Type locality: Europe (Pyrenees, Hungary, Austria).

*Calpe thalictri:* Treitschke, 1825: 168. Type locality: not stated.

*Calpe capucina* Graeser 1889: 260. Type locality: not stated.

*Calpe sodalis* Butler, 1878: 203 [27]. Type locality: Japan (Hokkaido, Yokohama).

*Calpe pallida* Schwingenschuss, 1938: 455 [28]. Type locality: Turkey.

*Calpe centralitalica* Dannehl, 1925: 12 [29]. Type locality: not stated.

*Calyptra thalictri* Sugi 1982: 861.

**Material examined:** 2 ♂, Ningxia, Jingyuan County, Liupan Mountain, Hongxia Forest Farm, 2010 m, 05.VII.2008, Lv Lin, 1 ♂, Hebei, Yanqing District, Dahaituo National Nature Reserve, 06.VIII.2006, Yang Zhaofu.

**Diagnosis:** Adult. (**Figure 2C**) Forewing length 38–48 mm in males and 39–49 mm in females. Male antennae bipectinate (0.75–0.8 mm). Head and thorax brown with grayish suffusion. Forewing light to dark brownish-grey with fine pink striations, transverse lines brown. Diagonal line predominantly distinct and brownish and frequently bordered with greyish-green on inner side, extending from pointed apex to the inner margin, merging at one-third to one-half the distance between lobe and hook. Reniform stigma light brown and diffuse, vertical bar faint. Subterminal line serrated, inner side with narrow brown band but broader posteriorly. Hindwing brown, with dark brown outer line and slightly infuscate posteriorly. Abdomen yellowish-gray and fusiform. Androtheca on male tibia II missing. Dorsum of male femur I normal.

**Male genitalia:** (**Figure 3B**). Uncus more robust and slightly shorter than in *C. gruesa*. Valvae symmetrically shaped, elongate, slightly tapering distally. Cucullus broad, corona with apex not extending to cucullus, stout and truncated. The saccular process digitiform and longer than in congeners. Juxta broad and shield-like and medial portion slightly convex. Aedeagus slender and straight, sparsely bearing minute spinule cornuti on the distal end of the vesica.

**Host plant:** *Thalictrum* spp. (Ranunculaceae).

**Distribution:** China (Heilongjiang, Liaoning, Xinjiang, Shandong, Henan, Zhejiang, Fujian, Sichuan, Yunnan, Ningxia, Hebei), Russia (Sakhalin, Lower Amur, Central Amur, Southern Kuriles (Kunashir and Shikotan Islands), Primorye, Southern Siberia, Western Siberia, Urals, European part, Northern Caucasus), Japan (Hokkaido, Honshu, Shikoku, Kyushu, Ryukyu Islands), Korea, Mongolia, Eastern Kazakhstan, Central Asia, Middle East, Ukraine, Central and Southern Europe, North Africa [30].

#### 3.1.3. *Calyptra hokkaida* (Wileman, 1922)

Figure 2E and Figure 3C.

*Calpe hokkaida* Wileman, 1922: 198. Type locality: Japan (Hokkaido).

*Calpe hoenei* Berio, 1956: 116. Type locality: China.

*Calyptra hokkaida* Bänziger 1983: 472.

**Material examined:** 2 ♂, Hubei, Shennongjia Forestry District, 28.VII.2006, Cai Lijun, Zhou Huifeng.

**Diagnosis:** Adult. (**Figure 2E**) Forewing length 43–50 mm in males, 48–53 mm in females. Head and thorax uniformly colored. Labial palps short slightly tufted, third segment barely perceptible. Male antennae bipectinate (0.12–0.14 mm). Forewing olive-green with subtle brownish markings and faint transverse stripes. Basal line dark brown, extending from costa towards inner margin. Diagonal line distinctive reddish-brown bordered with greenish line on inner side, merging at one-third to one-half the distance between lobe and hook. Antemedial line distinct, bordered with greenish band. Reniform stigma obscure faintly and delineated in brown. Subterminal line serrated, bordered with pale greenish suffusion, incurve near apex. Tornal hook distinct sometimes small. Antemarginal white spot missing. Hindwing pale brown, gradually darkening towards inner margin, with faint outer and subterminal lines. Hindwing fringes light yellowish. Androtheca on male tibia II present. Dorsum of male femur I normal.

**Male genitalia**: (**Figure 3C**). Uncus slightly arcuate, evenly tapering towards apex. Valvae asymmetrical, one valve U-shaped, the other cordate. Cucullus well rounded, setose, and expanded apically. Corona with apex extending to cucullus, slender, finger-like, and slightly inward-curved. The saccular process bifurcated, contributing to genital asymmetry. Juxta boat-shaped and convex medially with undulating apex and finely granulated surface. Aedeagus stout with some large, broadly conical cornuti. Vesica complex, with well-developed spines.

Host plant: Menispermaceae, less commonly Ranunculaceae and Papaveraceae (subfamily Fumarioideae).

**Distribution:** China (Hubei, Central–Eastern China), Russia (Sakhalin, Lower Amur, Central Amur, Primorye), Japan (Hokkaido), Korea [30].

#### 3.1.4. *Calyptra albivirgata* (Hampson, 1926)

Figure 2D and Figure 3D.

*Calpe albivirgata* Hampson, 1926: 373. Type locality: ‘’Omei Shan’’, [Emei Shan, Sichuan, W. China] Japan (Yokohama).

*Calyptra albivirgata* Sugi 1982.

**Material Examined:** 1 ♂, Gansu, Bigou Mountain King temple, Wen County, 10.VII,2002, Li Xiushan, Wang Jun.

**Diagnosis:** Adult. (**Figure 2D**). Forewing length 54–60 mm in males, 57–60 mm in females. Head and thorax reddish-brown. Male antennae bipectinate (0.37–0.4 mm), with elongated outer rami and short inner rami. Forewing reddish-brown with dark brown transverse stripes. Diagonal line distinct brownish and greyish, merging at one-third of the distance between lobe and hook and vestigial to obsolete posteriorly. Basal line dark brown, extending obliquely from costa to basal fold. Antemedial line dark brown, faint, slightly undulating, strongly inclined at base. Reniform stigma dark brown, narrow, elongated, indistinct. Medial line dark brown, broad, and diffuse, extending obliquely basad from the costa to lower angle of discal cell, with pronounced recurvature. Postmedial line blackish-brown, bordered externally with purplish-red. White spots in antemarginal variable, occasionally absent. Subterminal line dark brown. Outer margin medially arched, tornus forming acute tooth. Inner margin indentation with dentiform projection. Hindwings smoky brown with light yellowish fringes. Abdomen smoky brown and fusiform. Androtheca on male tibia II present. Dorsum of male femur I normal.

**Male genitalia:** (**Figure 3D**) Uncus prolonged, sparsely setose. Valvae apex blunt. Cucullus rounded, moderately setose. Corona prominent, dentate, arising more proximally than in congeners. Juxta horseshoe-shaped. Aedeagus apex bearing several robust spine-shaped cornuti rather than numerous minute ones. Genital morphology similar to *C. gruesa*, but saccular process narrower than that of latter species.

**Distribution:** China (Hunan, Sichuan, Gansu), Japan.

#### 3.1.5. *Calyptra orthograpta* (Butler, 1886)

Figure 2F,G, Figure 3E and Figure 4D.

*Calpe orthograpta* Butler, 1886: 25. Type locality: ‘’Darjiling’’ [Darjeeling, N. India].

*Calpe striata* Poujade, 1887: 139 [31]. Type locality: “Moupin” [Baoxing County, W. Sichuan, China].

*Calyptra* orthograpta Bänziger 1983: 478.

**Material examined:** 2 ♂, Sichuan, Qionglai City, Luohan Mountain, 06.VII.2005, Light trap; 1 ♀, Hubei, Enshi, Badong County, Lvcongpo Town, 12.VII.2006, Cai Lijun, Zhou Huifeng; 1 ♀, Hainan, Lingshui, Diaoluo Mountain, 930 m, 25.V.2008. Fu Qiang, 1♂, Hainan, Jianfengling, 950 m, 07.V.2008, Fu Qiang.

**Diagnosis:** Adult. (**Figure 2F,G**) Forewing length 46–57 mm in males, 48–58 mm in females. Male antennae unidentate, thorn-like tuberculi present near base and hair tuft less prominent than in *C. minuticornis.* Forewing greyish-brown or dark. Distinct brownish diagonal line bordered white outer line, merging directly at lobe. Basal line dark, slightly angled towards base. Medial line visible, inward-curved. Subterminal line serrated, bordered by faint yellowish suffusion along veins, more pronounced in fresh specimens. Reniform stigma typically appearing as narrow, diffuse vertical bar, occasionally terminating with white delineation. White spot in antemarginal area small but distinct. Hook at tornus missing. Hindwings more greyish than forewing. Fringes yellowish with dark outer border. Wings more elongate than in most congeners. Abdomen brownish. Androtheca (brush of scent scales) on male tibia II present. Dorsum of male femur with hair tuft.

**Male genitalia:** (**Figure 3E**) Uncus slender, hook-like, and strongly curved distally, with a pointed apex. Valvae broad and elongate, with a gently curved outer margin and dense, fine setae along the costal edge. Corona slender and indistinct, slightly inward-curved and tapering towards the apex. The saccular process elongated and rounded at apex, extending ventrally beyond the vinculum. Juxta broad, plate-like, and symmetrically structured. Aedeagus moderately elongated, slightly curved, and with a patch of weakly sclerotized, spine-like cornuti within the vesica.

**Female genitalia:** (**Figure 4D**) Papilla analis elongated, moderately sclerotized and densely covered with fine long setae mainly on lateral and apical margins. Apophysis anterior short, extending toward inner region of genitalia. Apophysis posterior more prominent and longer than apophysis anterior. Ostium bursae small, weakly sclerotized. Antrum less developed than in *C. gruesa* and *C. fletcheri*. Ductus bursae slender and slightly arcuate, moderately sclerotized. Absence of prominent ridges or hooks that differ from *C. gruesa* and *C. fletcheri.* Corpus bursa posterior side swollen and tumescent, anterior side saccate. Corpus bursae similar to that of *C. lata*, but with more diffuse sclerotization.

**Distribution:** China (Sichuan, Hubei, Hainan, Taiwan), India, Nepal, Japan, Thailand.

#### 3.1.6. *Calyptra lata* (Butler, 1881)

Figure 2I,J, Figure 3F and Figure 4A.

*Calpe lata* Butler, 1881: 21. Type locality: “Tokei” [Tokyo, Japan].

*Calpe aureola* Graeser, 1889: 260 [32]. Type locality: Russian Far East.

*Calyptra lata* Sugi 1982: 862.

**Material examined:** 1♂, Hebei, Yanqing District, Dahaituo National Nature Reserve, 06.VIII.2006, Yang Zhaofu; 1♀, Heilongjiang, Mudanjiang City, Mudan Peak, 02.VII.2006, Li Yan kai, Tan Jiangli.

**Diagnosis:** Adult. (**Figure 2I,J**) Forewing length 46–55 mm in males, 46–60 mm in females. Male antennae bipectinate (0.4 mm). Head, thorax, and abdomen gray-brown. Forewing yellowish-brown with purplish-red suffusion. Diagonal line distinct reddish-brown, merging at one-third to one-half the distance between lobe and hook. Basal, antemedial, and medial lines dark brown. Reniform stigma either absent or represented by one-to-two-minute black spots. Subterminal area with two dark undulating lines, manifesting as black dots along veins. Tornal hook pronounced. White spot in the antemarginal area absent. Hindwing pale yellowish-brown, with dark brown outer lines and terminal area. Hindwing fringes light yellow. Androtheca (brush of scent scales) present on male tibia II. Dorsum of male femur I normal.

**Male genitalia:** (**Figure 3F**) Uncus moderately elongated and slender, with uniform curvature. Cucullus rounded, corona with apex inward-curved and sparse setae. Sacculus narrow, less symmetrical at base. Saccular process digitiform and triangular. Juxta broad, sclerotized plate with a trapezoidal to shield-like shape. Aedeagus elongate, with less apical denticle, bearing only four to eight large edentate cornuti.

**Female genitalia:** (**Figure 4A**). Papila analis clearly defined membranous lobes. Apophyses slender and elongated, without sclerotized structures compared to congeners. Apophysis posterior slender and apophysis anterior short. Ostium bursae slightly sclerotized. Antrum neither excessively elongated nor complex. Ductus bursae relatively short and moderately sclerotized, without obvious curvature. Corpus bursae broadly rounded, predominantly membranous, basal region distinct sclerotized but lacking prominent ridges or protrusions.

**Host plants:** *Citrus* spp., *Corydalis* spp., *Thalictrum* spp.

**Distribution:** China (Heilongjiang, Hebei, Shandong, Fujian, Yunnan), Russia (Lower Amur, Primorye), Japan (Honshu, Kyushu), Korea [30].

#### 3.1.7. *Calyptra fletcheri* (Berio, 1956)

Figure 2H and Figure 4C.

*Calpe fletcheri* Berio, 1956: 118. Type locality: China.

*Calyptra fetcheri* Bänziger 1983: 472.

**Material examined:** 2 ♀, Hebei, Yanqing District, Dahaituo National Nature Reserve, 06.VII.2006, Yang Zhaofu.

**Description:** Adult. (**Figure 2H**) Forewing length 42–54 mm in males, 48–52 mm in females. Head compact, with well-defined forwardly projecting labial palpi. Male antennae bipectinate (0.3–0.48 mm). Forewing dark rufous tinge, triangular with slightly convex outer margin. Apex blunt rather than pointed. Dark brown venation apparent, creating pattern resembling dried leaves. Diagonal line from apex to inner margin distinctly dark brown, distally lined with red, merging at one-third to two-thirds of the distance between lobe and hook. Antemedial line blackish, basal and medial lines dark brown. Reniform stigma appearing as faint diffuse vertical bar. White spot in antemarginal area absent. Tornal hook small but distinct. Hindwing vibrant to forewing but slightly paler, with pronounced shading along the veins and margins. Fringes dark yellow. Thorax stout, densely scaled, concolorous with wing. Androtheca on male tibia II absent, dorsum of male femur I norma. Abdomen elongated cylindrical, tapering posteriorly, lighter brown.

**Female genitalia:** (**Figure 4C**) Papila analis lightly sclerotized with a delicate texture. Apophyses slender with moderate length. Apophysis posterior slender, sclerotized projections extending downward from the papillae. Apophysis anterior relatively slender and slightly curved. Ostium bursae relatively small. Antrum well defined, strongly sclerotized, extending toward ductus bursae. Ductus bursae moderately elongate and well sclerotized, gradually curved. Corpus bursae asymmetrical, with large, heavily sclerotized region forming ridge or lamellate structure.

**Distribution:** China (Hebei).

**Host Plants**: *Rubus ellipticus* (Rosaceae).

### 3.2. Key to the Species of Genus Calyptra

1. Androtheca present on male tibia II ....................................................................................... 2– Androtheca absent ...................................................................................................................... 62. Forewing reddish-brown; tornal hook distinct; white spots variable ……..... *C. albivirgata*– Forewing greyish or yellow-brown; other features diagnostic ............................................. 33. Forewing elongate; hindwing fringe yellow with dark outer edge ............. *C. orthograpta*– Forewing not elongate; fringe pale or uniform ....................................................................... 44. Diagonal line reddish-brown, distinct; tornal hook prominent ................................. *C. lata*– Diagonal line faint or curved; tornal hook reduced ............................................................... 55. Forewing olive green; diagonal line bordered greenish ....................................... *C. hokkaida*– Forewing greyish-brown with pink striations ......................................................... *C. thalictri*6. Forewing purplish-gray; yellowish subterminal spot near Cu1 ............................. *C. gruesa*– Forewing rufous; diagonal line dark, bordered red ............................................... *C. fletcheri*

### 3.3. World Checklist of Calyptra Ochsenheimer, 1816: 78

*C. albivirgata* (Hampson, 1926) (*Calpe*) W. China (Emei Shan, Sichuan); Japan (Yokohama) [33].*C. bicolor* (Moore, 1883) (*Calpe*) N.W. India (Punjab) [34].*C. eustrigata* (Hampson, 1926) (*Calpe*) Sri Lanka (Kandy) [33].*C. fasciata* (Moore, 1882) (*Calpe*) India (Shimla, Sikkim, Mumbai) [35,36].*C. fletcheri* (Berio, 1956) (*Calpe*) China [37].*C. gruesa* (Draudt, 1950) (*Calpe*) China (Taibai Shan, Shaanxi; West-Tianmu- Shan Zhejiang) [38].*C. hokkaida* (Wileman, 1922) (*Calpe*) Japan (Hokkaido) [39].*C. imperalis* (Grünberg, 1910) (*Calpe*) Rwanda (C. Africa) [40].*C. lata* (Butler, 1881) (*Calpe*) Japan (Tokyo) [41].*C. minuticornis* (Guenée, 1852) (*Calpe*) Indonesia (Java Island), India (Mumbai, Darjeeling), Sri Lanka [36,42].*C. nyei* Bänziger, 1979 India (Naga Hills) [43].*C. ophideroides* (Guenée, 1852) (*Calpe*) “Indes Orientales” (E. indies specifically, the regions of Southeast Asia, including Indonesia and surrounding areas) [42].*C. orthograpta* (Butler, 1886) (*Calpe*) India (Darjeeling) [44].*C. parva* Bänziger, 1979 India (Naga Hills, Manipur) [43].*C. pseudobicolor* Bänziger, 1979 India (Sikkim), Myanmar, Bhutan (Toungho) [43].*C. subnubila* (Prout, 1928) (*Calpe*) Indonesia (Mount Kerinci, Sumatra) [45].*C. thalictri* (Borkhausen, 1790) (*Phalaena*) Europe (Pyrenees, Hungary, and Austria) [46].*C. canadensis* (Bethune, 1865) (*Calpe*) North America (Canada) [47].

## 4. Discussion

This study represents a significant taxonomic revision of the genus *Calyptra* within Chinese fauna, including seven distinct species. Prior to this research, no comprehensive study had been conducted on *Calyptra* within China. By addressing critical gaps in previous studies, this research enhances our understanding of diversity in the genus.

The morphological analysis conducted in this study confirms the diagnostic value of several key characteristics for species identification within *Calyptra*, which is consistent with a previous study on the genus [14]. For example, the forewing span of *Calyptra* species ranges from 35 to 72 mm, with most species measuring between 50 mm and 60 mm. Smaller species have a forewing span of approximately 40–45 mm, while larger species range from 60 to 70 mm. Similarly, some key identification characteristics observed for species separation within this genus include variations in head and palpi coloration, which range from light to dark brown or grey, sometimes exhibiting subtle yellow, red, or green undertones. In *C. fletcheri*, the lateral surfaces of the head and palpi are distinctly red-tinged. Labial palpi are elongated and project anteriorly in a beak-like manner. The male antennal structure also exhibits remarkable interspecific variation in unidentate, unipectinate, or bipectinate forms, with pectinate forms showing gradually decreasing rami length from approximately the midpoint to the distal end, or at least up to 40% of the total antenna length. Rami length, ranging from 0.12 mm to 0.9 mm depending on the species, serves as a distinguishing morphological characteristic. Meanwhile, our findings indicate that male genitalia reveal significant structural diversity among Chinese *Calyptra* species, particularly in the configuration of the uncus, valvae morphology, and aedeagus structure.

The classification of *Calyptra* remained unclear until Duponchel [22] and Moore [35] provided key taxonomic insights in the mid-1800s. Berio [37] later revised the genus, synonymized four of nine species, and transferred *C. canadensis* into the genus *Percalpe*. Bänziger [1] expanded this revision, recognized 17 species, and treated *C. novaepommeraniae* as a subspecies of *C. minuticornis*, which was supported by inbreeding experiments. Based on Bänziger’s work, Zaspel and Branham [2] identified 18 species and reinstated *Percalpe* as a synonym of *Calyptra*, a conclusion later supported by molecular studies [48]. However, Bänziger argued against including *C. canadensis* in *Calyptra* due to differences in male genitalia, though his observations lacked molecular analysis. He also critiqued Zaspel and Branham’s [2] checklist and type locality, for instance, the misinterpretation of the *C. novaepommeraniae* type locality as India instead of Papua New Guinea. Despite these debates, this study follows Zaspel et al. [48] in recognizing 18 species and including *C. canadensis* within *Calyptra*, supported by molecular evidence.

Research on female genitalia of *Calyptra* remains limited, with a dissertation from the University of Florida [49] describing only five species, *C. albivirgata*, *C. orthograpta*, *C. pseudobicolor*, *C. subnubila*, and *C. eustrigata.* Historically, female genitalia were not considered crucial for classification, but this study describes *C. gruesa, C. fletcheri, C. orthograpta*, and *C. lata*, identifying valuable taxonomic features on female genitalia. Bänziger (personal communication) notes that China hosts the most diverse *Calyptra* fauna, particularly in Yunnan, Tibet, Guangdong, and Hainan, where species like *C. eustrigata, C. fasciata, C. nyei, C. ophideroides*, and *C. pseudobicolor* likely occur. Therefore, further research is necessary to explore *Calyptra* diversity, particularly within China, where taxonomic and phylogenetic studies remain insufficient.

## Figures and Tables

**Figure 1 insects-16-00534-f001:**
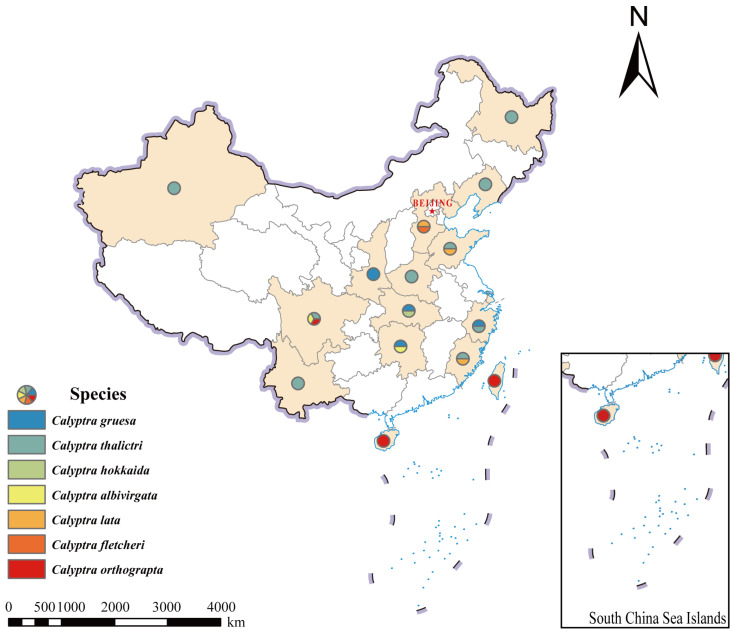
Geographical distribution of seven species of *Calyptra* recorded in Chinese fauna. Note: This map was created based on the standard map with approval number GS (2019)1653, downloaded from the website of the National Geomatics Center of China. The base map has not been modified.

**Figure 2 insects-16-00534-f002:**
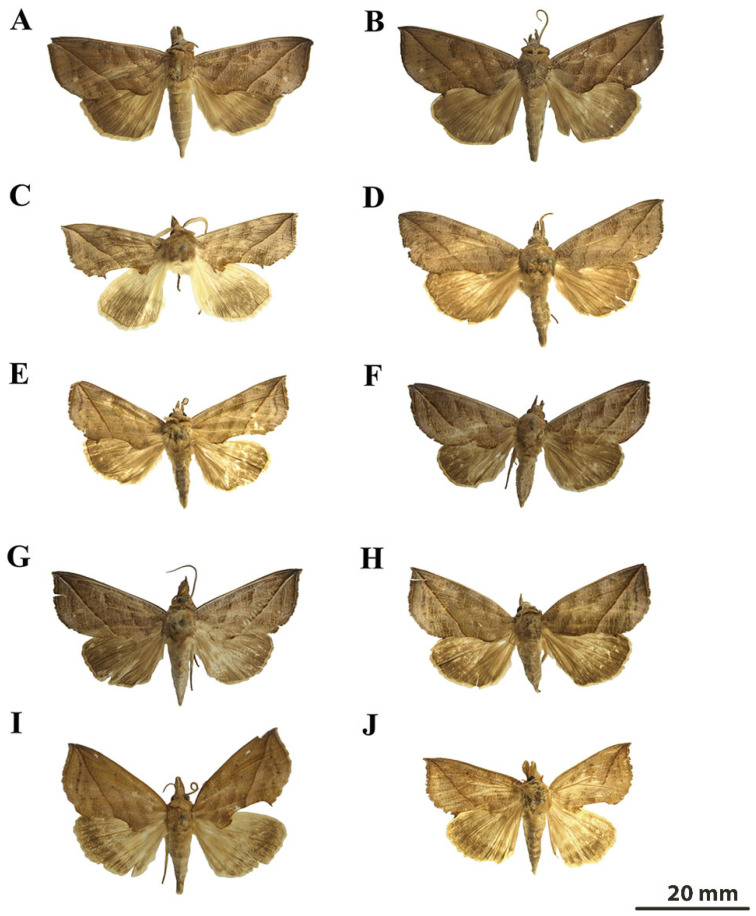
Adults of *Calyptra* spp.: (**A**) *C. gruesa* ♀; (**B**) *C. gruesa* ♂; (**C**) *C. thalictri* ♂; (**D**) *C. albivirgata* ♂; (**E**) *C. hokkaida* ♂; (**F**) *C. orthograpta* ♂; (**G**) *C. orthograpta* ♀; (**H**) *C. fletcheri* ♀; (**I**) *C. lata* ♂; (**J**) *C. lata* ♀.

**Figure 3 insects-16-00534-f003:**
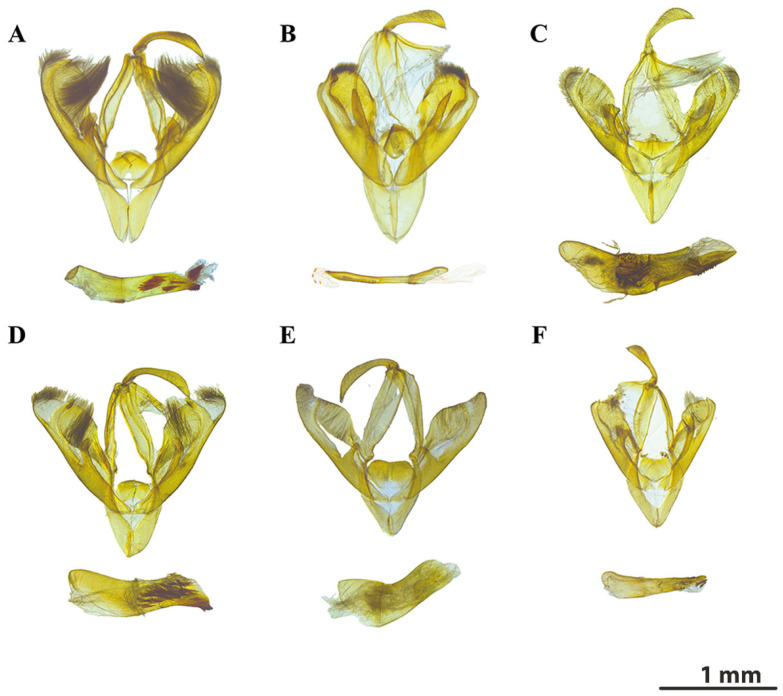
Male genitalia of *Calyptra* spp.: (**A**) *C. gruesa* (Slide No. LepiEreb001); (**B**) *C. thalictri* (Slide No. LepiEreb002); (**C**) *C. hokkaida* (Slide No. LepiEreb003); (**D**) *C. albivirgata* (Slide No. LepiEreb004); (**E**) *C. orthograpta* (Slide No. LepiEreb005); (**F**) *C. lata* (Slide No. LepiEreb10).

**Figure 4 insects-16-00534-f004:**
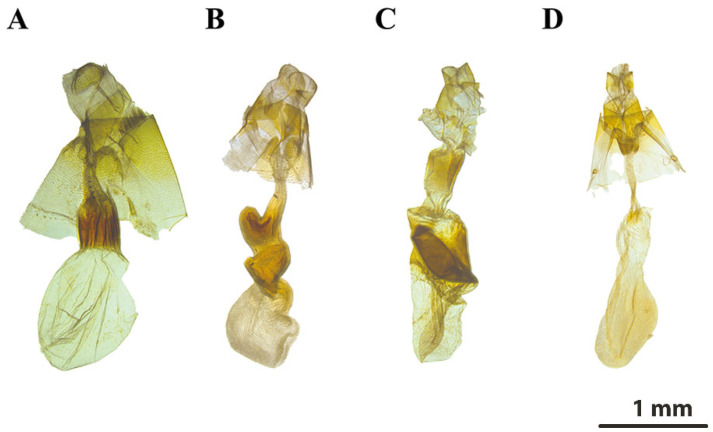
Female genitalia of *Calyptra* spp.: (**A**) *C. lata* (Slide No. LepiEreb007); (**B**) *Calyptra. gruesa* (Slide No. LepiEreb008); (**C**) *C. fletcheri* (Slide No. LepiEreb006); (**D**) *C. orthograpta* (Slide No. LepiEreb009).

## Data Availability

The specimens and extracted genitalia have been deposited in the Insects Systematics and Biodiversity Lab Entomological Museum, Northwest A&F University, Yangling, Shaanxi, China.

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
