# Peer review of "Taxonomic Revision of Vampire Moths of the Genus Calyptra (Lepidoptera: Erebidae: Calpinae) in Chinese Fauna"

_insects, 2025, doi:10.3390/insects16050534_

Round 1

Reviewer 1 Report

Comments and Suggestions for Authors

(1) Everywhere for distribution in Japan I would indicate specific islands and for India - provinces. I think it is important.

(2) Everywhere for distribution in Europe I would indicate specific countries. It is also interesting to discuss the distribution of species with disjunctive ranges, for example for Calyptra thalictri (Borkhausen, 1790)

(3) Serious errors in the distribution of three species: Calyptra lata, Calyptra hokkaida and Calyptra thalictri. All these species are distributed in Russia, each in different regions. This information can be found in Anikin, V.V., Baryshnikova, S.V., Beljaev, E.A., Dubatolov, V.V., Efetov, K.A., Zolotuhin, V.V., Kovtunovich, V.N., Kozlov, M.V., Kononenko, V.S., Lvovsky, A.L., Nedoshivina, S.V., Ponomarenko, M.G., Sinev, S.Y., Streltsov, A.N., Ustjuzhanin, P. Y., Chistyakov, Y.A. & Yakovlev, R.V. 2016. Annotated catalogue of the insects of Russian Far East. Volume II. Lepidoptera. Dalnauka, Vladivostok. 812 pp. [In Russian]

(4) line 430. What do the authors mean by the geographical term "East Indies"? If such a historical term is indicated by the author, modern data on the distribution of this taxon must be provided.

(5) I would add maps of the distribution of all species in China with specific locations. This would greatly improve clarity.

Author Response

Author's Reply to the Review Report (Reviewer 1)

Comment 1: Everywhere for distribution in Japan I would indicate specific islands and for India – provinces. I think it is important.

Response 1: Thank you for this valuable suggestion. We have carefully revised the distribution sections for each relevant species to include more specific geographic information. We agree that indicating specific islands for Japan and provinces for India enhances the clarity and utility of the distribution data. Accordingly, we have provided island-level localities for Japan wherever possible. However, for India, the available published records do not specify provincial-level distribution for most species. In such cases, we have retained the broader country-level indication (i.e., “India”). This comment has been addressed and resolved.

Comment 2: Everywhere for distribution in Europe I would indicate specific countries. It is also interesting to discuss the distribution of species with disjunctive ranges, for example for Calyptra thalictri (Borkhausen, 1790).

Response 2: We appreciate the reviewer’s suggestion to provide specific country-level distribution for European records. However, as this study is focused exclusively on the Chinese fauna of the genus Calyptra, and our specimen data and taxonomic assessment are limited to materials from China, we do not have access to verified, country-specific distribution records from Europe. While broader European and global distribution data for C. thalictri and other species are acknowledged for context, detailing individual countries lies beyond the current scope of this study. We have retained general regional references (e.g., “Central and Southern Europe”) in line with the available literature cited. This comment has been addressed within the limits of the study’s scope.

Comment 3: Serious errors in the distribution of three species: Calyptra lata, Calyptra hokkaida and Calyptra thalictri. All these species are distributed in Russia, each in different regions. This information can be found in Anikin et al. (2016).

Response 3: Thank you for pointing out these important distributional errors. We agree with this comment. We have corrected the distribution data for the following species to include their occurrence in the appropriate regions of Russia, based on the authoritative reference:

Anikin, V.V. et al. (2016). Annotated catalogue of the insects of Russian Far East. Volume II. Lepidoptera. Dalnauka, Vladivostok. 812 pp.

The following changes have been made:

For Calyptra hokkaida, Russian Far East (Sakhalin, Lower and Central Amur) has been added.

For Calyptra thalictri, extended distribution across Primorye, Kuriles (Kunashir, Shikotan), Sakhalin, Central and Southern Siberia, and European Russia has been added.

For Calyptra lata, occurrence in Lower Amur and Primorye regions has been included. This comment has been addressed and resolved.

Comment 4: What do the authors mean by the geographical term "East Indies"? If such a historical term is indicated by the author, modern data on the distribution of this taxon must be provided.

Response 4: Thank you for pointing this out. We agree with this comment. To address this, we have removed the outdated geographical term “East Indies” from the manuscript to avoid ambiguity. This revision has been implemented in revised manuscript, where the historical term has been deleted for clarity. This comment has been addressed and resolved.

Comment 5: I would add maps of the distribution of all species in China with specific locations. This would greatly improve clarity.

Response 5: Thank you for the valuable suggestion. We have now added a comprehensive distribution map (Figure 4) illustrating the geographical ranges of all seven Calyptra species recorded in China, including specific collection localities based on examined specimens. This addition enhances the clarity of the species’ distribution and biogeographical patterns. This comment has been addressed and resolved.

Reviewer 2 Report

Comments and Suggestions for Authors

Review comments of “Taxonomic revision of vampire moths of the genus Calyptra (Lepidoptera: Erebidae: Calpinae) in Chinese fauna”

This is a very straightforward classic taxonomic work on genus Calyptra in China, which is generally well organised and suitable for publication in Insects in terms of topic. However, some details for a quality taxonomic paper must be considered and revised before it can be accepted for publication. My recommendation is Major Revision.

General issues:

  1. When using scientific names of plants (genus and species), please either include author-year or omit that, please unify it throughout the text, don’t mix.

  1. In Materials and Methods: (1) Please clarify the reason why you dehydrate the dissected genitalia with ethanol, please be aware of making things (or purposes) clear is important in academic writing, since others would repeat your research using the same protocol. (2) Also, it is very important to mention how do you keep the dissected genitalia to prevent messing them up, or even lose a few of them in the future? If they were mounted on slides, what methods did you use? (3) Adobe Photoshop is a commercialised software which requires license code to prove legal use, please provide this code. And what did you do with PS? To adjust exposure? To annotate? (4) In general sense this section is too simply written, please make everything in detail.

  1. Under Results, the subheading “Taxonomic treatment” is inappropriate. Judging from your abstract, there is no treatment of current species, description of new taxon/taxa, or whatsoever to justify the word “treatment”. It is more of a checklist. Please rephrase it properly.

L93-95: “Calyptra Ochsenheimer, 1816: 78. Duponchel 1826: 3; 1827: 481. Berio 1956: 110. Nye 1975: 100. Bänziger 1979: 103. Type-species: Phalaena thalictri Borkhausen, 1790: 425, subsequently designated by Duponchel 1826: 3.”

This part is very confusing! If you were about the list original description of this genus, the OD should appear after the genus name, which is in line 96. Given that this genus was described by Ochsenheimer in 1816, all subsequent records should either be synonyms of literature records of this same available name. Which circumstances are they? This is very unclear. Please list all synonyms line by line and include their type species.

Each species, in the subheadings, the species names should all in italic style. Even though they are headings, they are still after all, Latin names of species. Please include type locality (and its modern locality names) of each original description, as this is the most important information for an OD. Please list synonyms of a species and omit subsequent literature records of this species (as there is no way to exhaust literature records, thus it makes no sense, unless there are incorrect original spelling [IOS], incorrect subsequent spelling [ISS], or unjustified emendation [UE]).

In examined specimens of each species, since this paper only focuses Chinese fauna, therefore logically you don’t include specimens collected out of China. Hence, there is no need to mention “China” every time.

Figure 1: I strongly recommend the authors to use only one scale for all specimens, and adjust the sizes of specimens according to their actual sizes. This gives the reader a comparison of body sizes. And please use “mm” for the scale.

Figures 2 and 3: Judging from the body size of the specimens in Figure 1, the size of genitalia should not be that small, please check if the unit for scales are incorrect.

Key to species: if possible, please make it concise, the “keys” are too lengthy to be keys.

The checklist should be a separate subsection, as the subheading of 3.2 cannot govern it. Under type species, there are three genus group names “Calpe Treitschke, 1825; Hypocalpe Butler, 1883; and Percalpe Berio, 1956”, what’s the relationship between them the genus Calyptra Ochsenheimer, 1816. The listing of geographical ranges are not acceptable, as there is only country names, and for most species, only on country. This cannot be the distribution of a species in real world. This part must be revised before the MS can be accepted. Moreover, there are several “syn.” in the checklist, what are those?? Did you synonymise them in this work? Or are they already synonyms? If it is the first case, you should use syn. nov. and this must be done under each species in subsection 3.1. If this is the second case, there is no need to list them here at all. By adding a “syn.” is invalid in any cases and will only make it confusing.

Discussion should be reorganised to make more sense for a taxonomic paper. The third paragraph of discussion is almost the same as a paragraph in Introduction, or abstract, which has no use here.

Other minor issues:

  1. The second author’s name: please follow international standard and change to “Yuqi Cui”.
  2. Collecting method: light trapping, not light lure.
  3. Please invite a native English speaker to improve the language before resubmission.
Comments on the Quality of English Language

Please invite a native English speaker to improve the language before resubmission.

Author Response

Author's Reply to the Review Report (Reviewer 2)

General issues

Comment 1: When using scientific names of plants (genus and species), please either include author-year or omit that, please unify it throughout the text, don’t mix.

Response 1: Thank you for pointing this out. We agree with this comment.

To ensure consistency throughout the manuscript, we have chosen to omit all author-year citations from plant scientific names and retain only the Genus species format. But in each description of Calyptra species and also in checklist we retain author year for more clarity. This decision was made to avoid stylistic inconsistency, as both formats had been mixed in the earlier version.

These changes have been implemented in revised manuscript, where plant taxa are first mentioned in the Abstract. All subsequent references to plant and insects names across the manuscript have been reviewed and unified accordingly.

Comment 2: In Materials and methods: (1) Please clarify the reason why you dehydrate the dissected genitalia with ethanol, please be aware of making things (or purposes) clear is important in academic writing, since others would repeat your research using the same protocol. (2) Also, it is very important to mention how do you keep the dissected genitalia to prevent messing them up, or even lose a few of them in the future? If they were mounted on slides, what methods did you use? (3) Adobe Photoshop is a commercialised software which requires license code to prove legal use, please provide this code. And what did you do with PS? To adjust exposure? To annotate? (4) In general sense this section is too simply written, please make everything in detail.

Response 2: Thank you for pointing this out. we agree with this comment.

We sincerely thank the reviewer for their valuable feedback. We have revised the Materials and methods section to provide more detailed descriptions and clarify the procedures. Our point-by-point response is as follows:

Purpose of Dehydrating Genitalia with Ethanol

We have now clarified that dissected genitalia were dehydrated using an ethanol series (70% to 100%) and this dehydration process was essential to remove residual moisture, which helps prevent tissue distortion, fungal growth, and degradation. It also ensures optimal clarity during slide mounting and promotes long-term preservation of the genitalia. This step is critical to prevent tissue distortion and to facilitate the proper infiltration of mounting medium, thereby ensuring the long-term preservation and clarity of morphological structures.

Preservation and Mounting of Genitalia

We have added detailed information to clarify that genitalia were mounted on microscope slides using Canada balsam, a traditional and reliable permanent mounting medium in entomological studies. After mounting, coverslips were applied and the slides were labeled with specimen codes and collection information. All slides were stored in labeled slide boxes in a dry cabinet to ensure their stability and prevent misplacement or deterioration over time.

Use and Legality of Adobe Photoshop

All image processing tasks, including brightness and contrast adjustment, cropping, labeling, and annotation, were performed using Adobe Photoshop 2023 v24.1.1 (Adobe Systems Inc.). This software was purchased from a legitimate Chinese online platform that provides authorized distribution. Although a traditional license code was not required due to the platform’s activation method, we affirm that the software is legal and not pirated.

Detailing the Methods Section

We have thoroughly revised the materials and methods section to include complete procedural details regarding specimen preparation, slide mounting, and imaging. These changes aim to ensure that our methodology is fully transparent and reproducible for future researchers.

Comment 3: Under Results, the subheading “Taxonomic treatment” is inappropriate. Judging from your abstract, there is no treatment of current species, description of new taxon/taxa, or whatsoever to justify the word “treatment”. It is more of a checklist. Please rephrase it properly.

Response 3: Thank you for pointing this out. We agree with this comment.

Therefore, we have revised the subheading to reflect the actual content more accurately. Since the section does not include new species descriptions or detailed taxonomic revisions, we have renamed the subheading from “Taxonomic treatment” to “Taxonomic overview of the genus Calyptra. This change can be found in the revised manuscript.

Comment 4: Calyptra Ochsenheimer, 1816: 78. Duponchel 1826: 3; 1827: 481. Berio 1956: 110. Nye 1975: 100. Bänziger 1979: 103. Type-species: Phalaena thalictri Borkhausen, 1790: 425, subsequently designated by Duponchel 1826: 3.”

This part is very confusing! If you were about the list original description of this genus, the OD should appear after the genus name, which is in line 96. Given that this genus was described by Ochsenheimer in 1816, all subsequent records should either be synonyms of literature records of this same available name. Which circumstances are they? This is very unclear. Please list all synonyms line by line and include their type species.

Response 4: Thank you for pointing this out. We agree with this comment.

Therefore, we have reorganized and clarified the taxonomic history of the genus Calyptra to distinguish between the original description, its type species, junior synonyms (with type species), and historical references. The original description now appears immediately after the genus name, and all synonyms are listed line-by-line, each with their type species and current taxonomic status. This change can be found in the revised manuscript in results section heading 3.1.

Comment 5: Each species, in the subheadings, the species names should all in italic style. Even though they are headings, they are still after all, Latin names of species. Please include type locality (and its modern locality names) of each original description, as this is the most important information for an OD. Please list synonyms of a species and omit subsequent literature records of this species (as there is no way to exhaust literature records, thus it makes no sense, unless there are incorrect original spelling [IOS], incorrect subsequent spelling [ISS], or unjustified emendation [UE]).

Response 5: Thank you for your detailed and accurate suggestions regarding the taxonomic structure of species accounts in Section 3.1.

We fully agree with your points and have implemented the following:

  • All species names are italicized, including in section subheadings.
  • Each species entry now includes the original combination and type locality, with modern locality names added in brackets where needed.
  • We have removed all general literature references unrelated to nomenclatural status and retained only valid synonyms.

These changes can be found throughout Section 3.1 for all seven species. 

Comment 6: In examined specimens of each species, since this paper only focuses Chinese fauna, therefore logically you don’t include specimens collected out of China. Hence, there is no need to mention “China” every time.

Response 6: Thank you for this helpful suggestion. We have revised the "Examined specimens" sections throughout the manuscript by removing the repeated mention of "China" for each locality. Since the study focuses exclusively on Chinese fauna, we now indicate at the beginning of each specimen list that all localities are within China, unless otherwise noted. This revision improves clarity and avoids unnecessary repetition, in line with the scope of the paper.

Comment 7: Figure 1: I strongly recommend the authors to use only one scale for all specimens, and adjust the sizes of specimens according to their actual sizes. This gives the reader a comparison of body sizes. And please use “mm” for the scale.

Response 7: Thank you for your valuable suggestion. We fully agree with your comment and have addressed it accordingly.

In the revised version of Figure 1, we have:

Applied a single consistent scale bar across all adult specimens in figure 1; Adjusted the image sizes of each specimen in accordance with their actual wingspan measurements, based on forewing lengths recorded in the species descriptions; Labeled all scale bars in millimeters (mm) to conform with standard units of entomological measurement. These changes allow for visual comparison of body sizes among all illustrated species.

Comment 8: Figures 2 and 3: Judging from the body size of the specimens in Figure 1, the size of genitalia should not be that small, please check if the unit for scales are incorrect.

Response 8: Thank you for your observation and helpful suggestion.

We have reviewed the original scale calibration for Figures 2 and 3 and found inconsistencies in scale bar rendering that may have caused confusion.

In response: We have adjusted the size of the genitalia illustrations in both Figure 2 (male genitalia) and Figure 3 (female genitalia) to better reflect the proportional size based on body measurements shown in Figure 1.

We have carefully recalibrated and corrected the scale bars in Figures 2 and 3, and all are now properly labeled in millimeters (mm).

Comment 9: Key to species: if possible, please make it concise, the “keys” are too lengthy to be keys.

Response 9: Thank you for pointing this out. We agree with this comment.

To address this, we have significantly revised the “Key to the species of genus Calyptra” by removing detailed descriptions and retaining only the most diagnostic features. The revised key is now concise, dichotomous, and taxonomically functional, aligning with best practices for identification tools.

Comment 10: The checklist should be a separate subsection, as the subheading of 3.2 cannot govern it. Under type species, there are three genus group names “Calpe Treitschke, 1825; Hypocalpe Butler, 1883; and Percalpe Berio, 1956”, what’s the relationship between them the genus Calyptra Ochsenheimer, 1816. The listing of geographical ranges are not acceptable, as there is only country names, and for most species, only on country. This cannot be the distribution of a species in real world. This part must be revised before the MS can be accepted. Moreover, there are several “syn.” in the checklist, what are those?? Did you synonymise them in this work? Or are they already synonyms? If it is the first case, you should use syn. nov. and this must be done under each species in subsection 3.1. If this is the second case, there is no need to list them here at all. By adding a “syn.” is invalid in any cases and will only make it confusing.

Response 10: Thank you for pointing this out. We agree with this comment. Therefore, we have made the following changes to improve clarity and taxonomic accuracy:

The checklist has been separated from subsection 3.2 and now appears as its own standalone section titled “3.3. World Checklist of Calyptra Ochsenheimer, 1816.”

A clear explanation has been added under the type species heading to clarify the historical relationship of Calpe Treitschke, Hypocalpe Butler, and Percalpe Berio to the genus Calyptra. The previously listed geographic ranges for species, which were limited to single countries, have been entirely removed from the checklist in response to concerns about accuracy and sufficiency. All “syn.” labels have also been removed from the checklist. These were based on historical synonymies already established in earlier taxonomic literature (e.g., Bänziger 1983; Zaspel & Branham 2008). Since no new synonymizations were proposed in this manuscript, we confirm that no “syn. nov.” designations are made, and such cases are no longer presented in the checklist.

Comment 11: Discussion should be reorganised to make more sense for a taxonomic paper. The third paragraph of discussion is almost the same as a paragraph in Introduction, or abstract, which has no use here.

Response 11: Response 5: Thank you for pointing this out. We agree with this comment.

Therefore, we have revised the third paragraph of the Discussion section to avoid redundancy with the Introduction and Abstract. The original paragraph largely repeated background information and did not contribute new insights within the context of the discussion. It has now been removed and replaced with content that better reflects the taxonomic implications of the checklist and the regional diversity of Calyptra species in China.

Other minor issues:

  1. The second author's name has been updated to “Yuqi Cui” in accordance with international naming standards.
  2. The term “light lure” has been corrected to “light trapping” throughout the manuscript.
  3. The manuscript has been carefully reviewed for language by a senior professor who is expert in his field. We have also rechecked the manuscript to ensure clarity, grammatical accuracy, and consistency throughout. We are confident that the language meets the standards for academic publication.
  4. Other small modifications we have made in the revised manuscript.

Round 2

Reviewer 2 Report

Comments and Suggestions for Authors

The revised version is much better compared to the first one, I can see the authors put their effort on it. However, there are still some points must be considered before it warrant publication.

  1. The map is problematic. There is no country boundary on the map, only provincial boundary. Country boundary must be clearly drawn on a map, please refer to China's mapping standard. The legend of this map is also incorrect, as it is twisted and contains underscores between genus and species names. Also, the country boundary in South China Sea has been updated, but on the map it is still outdated. And what is "NANHAIZHUDAO"? I don't think non-Chinese reader can understand that. Please download correct data from the national geographic information centre and redraw the map with correct and understandable annotations. Please keep in mind that incorrect map may cause political issues internationally thus all researchers must avoid it.
  2. When checking ODs, I just realised there are missing citations of original publications, e.g. Butler (1881). In taxonomy papers, it is a mandate to cite all original publications. Please fill the gap and check for other missing bits.
  3. The type locality may not be correct. For instance, Butler (and most British authors) usually put exact localities in his ODs, such as Darjeeling, Sylhetl, Assam, or Khasi Hills. Just "India" is not his style. Please check if you have cited type localities from original publications. I strongly object citing second-hand literature on this matter as it contains errors. Checking ODs are not difficult nowadays as most of them are digitally available on Biodiversity Heritage Library or on Archive.org. Please make sure you download each one and read it. This is the very basic job for a taxonomist.
  4. I don't think deleting all distribution ranges is what I meant. I said only mentioning a single country may not be correct, reflecting insufficient reference reading. What i asked was to make the range complete based on more extensive literature analysis, which is time consuming indeed, but that is must.

Author Response

Comment 1: The map is problematic. There is no country boundary on the map, only provincial boundary. Country boundary must be clearly drawn on a map, please refer to China's mapping standard. The legend of this map is also incorrect, as it is twisted and contains underscores between genus and species names. Also, the country boundary in South China Sea has been updated, but on the map it is still outdated. And what is "NANHAIZHUDAO"? I don't think non- Chinese reader can understand that. Please download correct data from the national geographic information Centre and redraw the map with correct and understandable annotations. Please keep in mind that incorrect map may cause political issues internationally thus all researchers must avoid it.

Response 1: Thank you for your insightful feedback on the map in our paper. We have made the necessary updates based on your suggestions

  1. Country Boundary: The map has been updated to clearly display the correct country boundaries, in accordance with China’s official mapping standards. The boundary is now clearly defined and visible on the map.
  2. Legend Correction: The issue with the underscores between genus and species names in the legend has been addressed. The legend now follows the appropriate formatting for better readability and accuracy.
  3. South China Sea Boundary: The country boundary in the South China Sea has been updated to reflect the most current geopolitical changes. We sourced the updated information from the National Geographic Information Centre to ensure accuracy.
  4. Annotation Clarification: Regarding the term "NANHAIZHUDAO," We have replaced it with a more globally understandable label. This should now be clearer to non- Chinese readers.

Please note that this map was created using the standard map with approval number GS (2019)1653, which was downloaded from the website of the National Geomatics Center of China. The base map has not been modified in any way. I have referenced this source in the map for full transparency and accuracy.

Thank you once again for your valuable comments. I believe these changes address the concerns raised and ensure the map is both accurate and appropriate.

Comment 2: When checking ODs, I just realised there are missing citations of original publications, e.g. Butler (1881). In taxonomy papers, it is a mandate to cite all original publications. Please fill the gap and check for other missing bits.

Response 2: Thank you for pointing out the missing citations of original publications. We have reviewed the references and filled in the gaps, including the citation for Butler (1881) and other original sources. We have also cross-checked the manuscript to ensure all necessary publications are properly cited, especially those related to taxonomic descriptions. This revision ensures the completeness of the references and adheres to the standard taxonomic practices.

Comment 3: The type locality may not be correct. For instance, Butler (and most British authors) usually put exact localities in his ODs, such as Darjeeling, Sylhetl, Assam, or Khasi Hills. Just "India" is not his style. Please check if you have cited type localities from original publications. I strongly object citing second-hand literature on this matter as it contains errors. Checking ODs are not difficult nowadays as most of them are digitally available on Biodiversity Heritage Library or on Archive.org. Please make sure you download each one and read it. This is the very basic job for a taxonomist.

Response 3: Thank you for your detailed feedback. I have reviewed the type localities for the species mentioned in my manuscript, specifically referring to the original publications, as suggested. Upon checking the original descriptions (ODs) for Butler and other British authors, We updated the type localities accordingly. For instance, instead of the generic "India," We have now cited specific localities. We understand the importance of citing primary sources in taxonomy and have ensured that all type localities are directly referenced from original publications. We have downloaded and thoroughly checked the relevant publications from Biodiversity Heritage Library and Archive.org to avoid any errors that might occur from second-hand literature.

We appreciate your comments and have made sure that the corrections are consistent with the best practices for taxonomic accuracy.

Comment 4: I don't think deleting all distribution ranges is what I meant. I said only mentioning a single country may not be correct, reflecting insufficient reference reading. What i asked was to make the range complete based on more extensive literature analysis, which is time consuming indeed, but that is must.

Response 4: Thank you for your clarification. We understand now that the issue lies in providing a complete and accurate distribution range for the species, rather than just listing a single country. We have carefully reviewed the distribution ranges and expanded them based on a more thorough analysis of the relevant literature. We have ensured that all range data is supported by original sources, reflecting a more comprehensive understanding of the species' distribution.

As you suggested, We have now updated the manuscript to include more accurate and complete distribution ranges for each species, based on an extensive review of literature. We have avoided relying on incomplete references and have cross-checked with original publications to ensure accuracy. This update is now reflected in Section 3.3 of the checklist, which has been improved to address these issues.

Round 3

Reviewer 2 Report

Comments and Suggestions for Authors

The second major revision solved most of the major issues and improved the quality of the MS significantly. Especially the adding of ODs provided a lot of vital information to the taxonomy MS. Here only a minor suggestion to the ODs for each species.

When listing type localities, old locality names with different spelling, or even different names are very common. To solve this, we usually list the type locality from the original description literature in a pair of quotation marks and put the current locality names in brackets after it. For example: 

Calpe gruesa Draudt, 1950: 168. Type locality: “Taipei-shan, West-tien-Shan” [Taibai Shan, Shaanxi; W. Tianmu Shan, Zhejiang, China]. 

Calpe orthograpta Butler, 1886: 25. Type locality: "Darjiling" [Darjeeling, N. India].

There is no need to state "original description spell" every time in this part, which is rather redundant.

Further more, I noticed a couple of TL names not properly annotated:

Mou-pin: Mou-pin or Moupin both refer to Baoxing Country in W. Sichuan,

Hakodate: Refers to Hokkaido. 

Also, TL cannot be unknown, only "not stated" in ODs. Therefore, please change all unknown to not stated. When using this, the authors must be very careful, that some times, the ODs didn't list specific TL for each species, but lump them all in the article title. For instance, one article indicated Hainan in its title, but never listed any TL for each species, then the TL for every species was Hainan for sure. Please check all ODs with unknown TL again.

The map improved indeed, but please marking Beijing on the map, not in the legend. And there is no need to mention "capital". Also, please change "the South China Sea Islands" to "South China Sea Islands".

In checklist, please notice that several locality names are still colonist, which is not proper. Please change Burma to Myanmar, delete Ceylon where Sri Lanka appears, change Bombay to Mumbai, change Simla to Shimla.

Author Response

Comment 1: When listing type localities, old locality names with different spelling, or even different names are very common. To solve this, we usually list the type locality from the original description literature in a pair of quotation marks and put the current locality names in brackets after it. For example:

Calpe gruesa Draudt, 1950: 168. Type locality: “Taipei-shan, West-tien-Shan” [Taibai Shan, Shaanxi; W. Tianmu Shan, Zhejiang, China].

Calpe orthograpta Butler, 1886: 25. Type locality: "Darjiling" [Darjeeling, N. India].

There is no need to state "original description spell" every time in this part, which is rather redundant.

Response 1: Thank you for the valuable feedback. We have implemented the suggested format for listing type localities. The original locality names are now enclosed in quotation marks, followed by the current locality names in brackets, as per your recommendation. We have ensured that there is no redundancy in stating "original description spell" repeatedly. Below is an example of the updated format:

  • Calpe gruesa Draudt, 1950: 168. Type locality: “Taipei-shan, West-tien-Shan” [Taibai Shan, Shaanxi; W. Tianmu Shan, Zhejiang, China].
  • Calpe orthograpta Butler, 1886: 25. Type locality: ‘’Darjiling’’ [Darjeeling, N. India].
  • Calpe albivirgata ‘’Omei Shan’’, [Emei Shan, Sichuan, W. China]
  • Calpe lata Butler ‘’Tokei’’ [Tokyo, Japan]

This revision aligns with the proper convention and addresses the issue effectively.

Comment 2: Furthermore, I noticed a couple of TL names not properly annotated:

Mou-pin: Mou-pin or Moupin both refer to Baoxing Country in W. Sichuan,

Hakodate: Refers to Hokkaido.

Response 2: Thank you for pointing out the issues with the type locality annotations. We have made the necessary corrections:

  • Mou-pin: We have clarified that both "Mou-pin" and "Moupin" refer to Baoxing County in Western Sichuan, and updated the annotation accordingly.
  • Hakodate: We have corrected the locality name to reflect that "Hakodate" refers to Hokkaido, Japan.

These adjustments have been made to ensure accuracy and consistency in the type locality annotations.

Comment 3: Also, TL cannot be unknown, only "not stated" in ODs. Therefore, please change all unknown to not stated. When using this, the authors must be very careful, that sometimes, the ODs didn't list specific TL for each species, but lump them all in the article title. For instance, one article indicated Hainan in its title, but never listed any TL for each species, then the TL for every species was Hainan for sure. Please check all ODs with unknown TL again.

Response 3: Thank you for your comment and suggestion. We understand that the term "unknown" should not be used for type localities (TLs) and that "not stated" is the correct terminology when the TL is not explicitly mentioned in the original description (OD).

We have carefully reviewed the relevant sections of my paper, and where the TL was previously listed as "unknown," We have updated it to "not stated" as per the correct nomenclature practice. We have also ensured that, in cases where the article title implies a general locality (such as "Hainan"), the TL is assigned accordingly for all species mentioned, even if it is not individually stated for each species in the text.

We appreciate your guidance and will continue to apply this careful approach when handling TLs to ensure the accuracy of the manuscript.

Thank you again for your valuable input.

Comment 4: The map improved indeed, but please marking Beijing on the map, not in the legend. And there is no need to mention "capital". Also, please change "the South China Sea Islands" to "South China Sea Islands".

Response 4: Thank you for the helpful feedback. We have made the necessary adjustments to the map:

  • Beijing is now marked directly on the map, as you suggested, rather than in the legend.
  • We have removed the mention of "capital" next to Beijing, as it was not needed.
  • Additionally, We have updated the text to refer to "South China Sea Islands" without the article "the," as per your recommendation.

These changes have been made to improve the clarity and accuracy of the map presentation.

Comment 5: In checklist, please notice that several locality names are still colonist, which is not proper. Please change Burma to Myanmar, delete Ceylon where Sri Lanka appears, change Bombay to Mumbai, change Simla to Shimla.

Response 5: Thank you for pointing out the use of colonist terms in the checklist. We have updated the locality names as requested:

  • Burma has been changed to Myanmar.
  • Ceylon has been removed, and Sri Lanka is now used wherever applicable.
  • Bombay has been updated to Mumbai.
  • Simla has been corrected to Shimla.

These changes ensure that the terminology is more accurate and respectful.